# Unveiling the circRNA-Mediated Immune Responses of Western Honey Bee Larvae to *Ascosphaera apis* Invasion

**DOI:** 10.3390/ijms24010613

**Published:** 2022-12-29

**Authors:** Yaping Ye, Xiaoxue Fan, Zongbing Cai, Ying Wu, Wende Zhang, Haodong Zhao, Sijia Guo, Peilin Feng, Qiming Li, Peiyuan Zou, Mengjun Chen, Nian Fan, Dafu Chen, Rui Guo

**Affiliations:** 1College of Animal Sciences (College of Bee Science), Fujian Agriculture and Forestry University, Fuzhou 350002, China; 2Apitherapy Research Institute of Fujian Province, Fuzhou 350002, China

**Keywords:** non-coding RNA, circRNA, honey bee, *Apis mellifera*, *Ascosphaera apis*, larva, immune response, regulatory network

## Abstract

Western honey bee (*Apis mellifera*), a eusocial insect with a superior economic and ecological value, is widely used in the beekeeping industry throughout the world. As a new class of non-coding RNAs (ncRNAs), circular RNAs (circRNAs) participate in the modulation of considerable biological processes, such as the immune response via diverse manners. Here, the identification, characteristic investigation, and molecular verification of circRNAs in the *Apis mellifera ligustica* larval guts were conducted, and the expression pattern of larval circRNAs during the *Ascosphaera apis* infection was analyzed, followed by the exploration of the potential regulatory part of differentially expressed circRNAs (DEcircRNAs) in host immune responses. A total of 2083 circRNAs in the larval guts of *A. m. ligustcia* were identified, with a length distribution ranging from 106 nt to 92,798 nt. Among these, exonic circRNAs were the most abundant type and LG1 was the most distributed chromosome. Additionally, 25, 14, and 30 up-regulated circRNAs as well as 26, 25, and 62 down-regulated ones were identified in the *A. apis*-inoculated 4-, 5-, and 6-day-old larval guts in comparison with the corresponding un-inoculated larval guts. These DEcircRNAs were predicted to target 35, 70, and 129 source genes, which were relative to 12, 23, and 20 GO terms as well as 11, 10, and 27 KEGG pathways, including 5 cellular and humoral immune pathways containing apoptosis, autophagy, endocytosis, MAPK, Toll, and Imd signaling pathways. Furthermore, complex competing endogenous RNA (ceRNA) regulatory networks were detected to be formed among DEcircRNAs, DEmiRNAs, and DEmRNAs. The Target DEmRNAs were engaged in 24, 20, and 25 functional terms as well as 62, 80, and 159 pathways, including several vital immune defense-associated pathways, namely the lysosome, endocytosis, phagosome, autophagy, apoptosis, MAPK, Jak-STAT, Toll, and Imd signaling pathways. Finally, back-splicing sites within 15 circRNAs and the difference in the 9 DEcircRNAs’ expression between un-inoculated and *A. apis*-inoculated larval guts were confirmed utilizing molecular methods. These findings not only enrich our understanding of bee host-fungal pathogen interactions, but also lay a foundation for illuminating the mechanism underlying the DEcircRNA-mediated immune defense of *A. m. ligustica* larvae against *A. apis* invasion.

## 1. Introduction

Honey bees not only pollinate a substantial quantity of wild flowers and crops but also produce an array of API products, such as honey, royal jelly, propolis, bee pollen, and beeswax, thus playing essential ecological and economical roles. As a kind of eusocial insect, honey bees are prone to infections by various bacteria, fungi, and viruses [1]. Among these, *Ascosphaera apis* is a representative fungal pathogen that exclusively infects bee larvae and triggers chalkbrood, a widespread fungal disease that causes sharp reductions in the adult bee population and colony productivity [2]. The western honey bee (*Apis mellifera*) is an excellent bee species that has a series of advantageous adaptions, including strong oviposition and foraging abilities, and is widely used in beekeeping practices across the world [3]. However, *A. mellifera* larvae are susceptible to an *A. apis* infection and chalkbrood, resulting in losses among reared *A. mellifera* colonies [4]. Recent studies have shown that chalkbrood is on the rise due to the rapid development of the worldwide trade of API products [5].

Different from other linear transcripts, circRNAs are circular RNA molecules formed by the reverse cleavage of precursor mRNAs (pre-mRNAs) that lack the 5′ cap and 3′ tail, hence, they are able to tolerate the digestion by exonucleases. Their covalently closed ring structures allow circRNAs to have diverse modes of action, e.g., circRNAs with miRNA response elements (MREs) can indirectly regulate the expression of downstream target genes by base pairing with target miRNAs [6]; some circRNAs exert biological functions by modulating the transcription of source genes [7]. Increasing numbers of studies have demonstrated that circRNAs are capable of regulating a suite of life activities, such as neuronal development [8], immune defense [9], and pathogen infection [10]. For example, Li et al. [11] found that the NF90/NF110 released from circRNAs could bind to viral mRNAs to regulate antiviral immune responses and further implicate the coordinated regulatory functions of the corresponding circRNAs. Recent evidence suggested that some circRNAs containing internal ribosome entry sites (IRESs) and/or N^6^-methyladenosine (m6A) modifications are able to encode small peptides or proteins to exert functions in several biological processes of a great importance, such as the immune defense as well as disease occurrence and progression [12,13]. However, only circRNAs in a few model species, e.g., humans [14], *Arabidopsis thaliana* [15], and *Drosophila* [16], have been extensively investigated, so limited progress has been made regarding circRNAs in insects, including honey bees, so far. For example, Thölken et al. [17] identified 1263 circRNAs in the brains of *A. mellifera* nurse and forager bees by using a linear RNA-digested strategy-based deep sequencing and detected that the expression level of circAm*rsmep2* was accumulated with the increasing age of the bees, while the level of circAm*rad* appeared to be both age-independent and correlated with the role of the bees.

Thanks to the rapid development and continuous breakthrough of deep sequencing technologies and the associated bioinformatics, an increasing number of circRNAs have been discovered in insects, including *Drosophila* [18], *Bombyx mori* [19], and *Laodelphax striatellus* [20]. For instance, Hu et al. [19] observed that a total of 3638 circRNAs in the midguts of *B. mori* were differentially expressed after a BmCPV infection and the source genes of the differentially expressed circRNAs (DEcircRNAs) were engaged in some crucial pathways, e.g., the stress response, peroxidase activity, and oxidoreductase. Previously, our team investigated the differential expression profiles and regulatory networks of circRNAs in the midguts of *Apis cerana cerana* workers, identified 494 DEcircRNAs in the guts of 6-day-old *A. c. cerana* larvae infected with *A. apis*, and found that some source genes of the DEcircRNAs were involved in oxidative phosphorylation, as well as cellular- and humoral-immune pathways [21]. Recently, we revealed that ame_circ_000115 participated in the response of *Apis mellifera ligustica* to an *A. apis* invasion based on an RNAi-based functional investigation [22].

It has been suggested that circRNAs modulate the interactions between insects and pathogens or pesticides [23]. Feng et al. [24] reported that *circ1-3p* in *Tetranychus cinnabarinus* competitively linked to miR-1-3p to regulate the expression of the target gene *TcGSTm04*, thereby further affecting the sensitivity of *T. cinnabarinus* to cyflumetofen. Compared to a few other model insects, such as *Drosophila* and *B. mori*, studies on circRNA-mediated bee–pathogen interactions were very limited. More recently, our group conducted an in-depth investigation of the expression patterns and regulatory networks of circRNAs in the midguts of both *A. m. ligustica* and *A. c. cerana* worker bees in response to the *Nosema ceranae* infection, and further deciphered the diverse regulatory roles of DEcircRNAs in the host responses [25,26]. However, whether circRNAs are involved in the responses of *A. m. ligustica* larvae to an *A. apis* infection is completely unknown at this present stage.

In this current work, on the basis of our previously obtained high-quality transcriptome data, circRNAs in the guts of *A. m. ligustica* larvae were characterized, followed by the molecular validation of back-splicing sites. Additionally, the differential expression patterns of circRNAs were surveyed and the regulatory manners of DEcircRNAs were investigated. Further, the DEcircRNA-DEmiRNA-DEmRNA regulatory networks associated with host immune defenses were constructed and analyzed. Finally, the differential expression of DEcircRNAs was verified using RT-qPCR. To our knowledge, this is the first report on the circRNA-mediated responses of *A. m. ligustica* larvae to an *A. apis* infection. The findings from this work could provide not only a basis for unveiling the mechanisms underlying the DEcircRNA-regulated responses of *A. m. ligustica* larvae to an *A. apis* invasion but also new insights into host-pathogen interactions during chalkbrood disease.

## 2. Result

### 2.1. Quantity and Characteristics of A. m. ligustica circRNAs

In total, 7,471,892, 6,905,870, 8,328,540, 7,893,288, 7,350,996, and 5,845,700 anchor reads were identified in the AmCK1, AmCK2, AmCK3, AmT1, AmT2, and AmT3 groups, respectively (Appendix A). Then, 3,160,593, 3,734,903, 4,154,993, 3,862,442, 3,826,332, and 3,040,217 anchor reads were, respectively, mapped to the *A. mellifera* reference genome (Appendix A). Additionally, 1006, 1116, 1198, 1273, 1090, and 1086 circRNAs were identified in the six groups, respectively. After removing the redundant circRNAs, a total of 2083 circRNAs were discovered, with a length distribution ranging from 106 nt to 92,798 nt (Figure 1A). Moreover, annotated exonic circRNAs (64.91%) were the most abundant, followed by single exonic circRNAs (16.42%) and exonic intronic circRNAs (7.73%), as shown in Figure 1B. LG1 (13.43%) was the most distributed chromosome by circRNAs, followed by LG2 (9.00%) and LG8 (8.24%), while LG16 (3.07%) was enriched by the fewest circRNAs (Figure 1C).

### 2.2. Analysis of the DEcircRNAs Engaged in Host Responses to A. apis Invasion

In the AmCK1 vs. AmT1, AmCK2 vs. AmT2, and AmCK3 vs. AmT3 comparison groups, 25, 14, and 14 upregulated circRNAs as well as 26, 25, and 25 downregulated circRNAs were detected, respectively (Figure 2A). The Venn analysis showed that there was no shared DEcircRNAs among these three comparison groups, whereas the numbers of unique DEcircRNAs were 46, 35, and 87, respectively (Figure 2B).

### 2.3. Investigation and Annotation of the Source Genes of DEcircRNAs

The DEcircRNAs in the AmCK1 vs. AmT1 comparison group were predicted to target 35 source genes, which were involved in seven biological process-associated GO terms (cellular processes, metabolic processes, and biological regulation, etc.), two cellular component-associated terms (membrane parts and membranes), and three molecular function-associated terms (the binding, catalytic, and transporter activities) (Figure 3A). The DEcircRNAs in the AmCK2 vs. AmT2 comparison group were found to target 70 source genes, which were engaged in 23 functional terms, such as the cellular process, binding, and cell part (Figure 3B). The DEcircRNAs in the AmCK3 vs. AmT3 comparison group were predicted to target 129 source genes, which were related to 20 GO terms, including the metabolic processes, cell parts, and catalytic activity (Figure 3C). In addition, the annotation of the KEGG database showed that the source genes in the AmCK1 vs. AmT1 comparison group were relevant to 11 KEGG pathways, including apoptosis, a cellular immune pathway (Figure 3D). The source genes in the AmCK2 vs. AmT2 comparison group were related to 10 pathways, also including apoptosis (Figure 3E), whereas the source genes in the AmCK3 vs. AmT3 comparison group were associated with 27 pathways, including 5 immune pathways such as the apoptosis, autophagy, endocytosis, MAPK, Toll, and Imd signaling pathways. (Figure 3F). The numbers of source genes involved in cellular and humoral immune pathways are summarized in Figure 3G.

### 2.4. Analysis of DEcircRNA-Involved ceRNA Regulatory Networks

CeRNA regulatory network analysis indicated that 21, 6, and 21 DEcircRNAs in the three comparison groups could bind to 10, 3, and 7 DEmiRNAs and further target 184, 94, and 195 DEmRNAs, respectively (Figure 4). Further investigation suggested that a subseries of DEcircRNAs could target multiple DEmiRNAs simultaneously, e.g., novel_circ_000161 in the 6-day-old comparison group could target both miR-6046 and miR-4488-y. Additionally, some DEmiRNAs could also be targeted by several DEcircRNAs at the same time, e.g., miR-8440-y could be targeted by 2, 4, and 8 DEcircRNAs in the aforementioned three comparison groups.

Furthermore, targets in the 4-day-old comparison group could be annotated to 24 GO terms, such as single-organism processes, cellular component organization or biosynthesis, macromolecular complexes (Figure 5), and 62 KEGG pathways (Figure 6A), including 3 immune pathways (lysosome, endocytosis, and MAPK signaling pathways). Targets in the AmCK2 vs. AmT2 comparison group were involved in 10 biological process-associated terms, 5 cellular component-associated terms, and 5 molecular function-associated terms (Figure 5), as well as 80 KEGG pathways (Figure 6B), including 4 immune pathways (lysosome, phagosome, MAPK, and Jak-STAT signaling pathways). Targets in the AmCK3 vs. AmT3 comparison group were relevant to 25 GO terms, such as the binding and signal transducer activity (Figure 5), and 159 KEGG pathways (Figure 6C), including 7 immune pathways (i.e., the lysosome, endocytosis, phagosome, autophagy, apoptosis, and MAPK, Jak-STAT, Toll, and Imd signaling pathways). Detailed information about the immune pathway-related targets is summarized in Appendix A.

### 2.5. Verification of Back-Splicing Sites within circRNAs

A PCR amplification of 15 randomly selected circRNAs was conducted using the corresponding divergent primers, and the Sanger sequencing of the amplified products confirmed the authenticity of the back-splicing sites and the reliability of our transcriptome data (Figure 7).

### 2.6. Validation of DEcircRNAs by RT-qPCR

Our RT-qPCR detection suggested that the differences of nine randomly selected DEcircRNAs between un-inoculated and *A. apis*-inoculated larval guts were roughly the same as those in the sequencing data (Figure 8), further validating the reliability of the transcriptome datasets used in this work.

## 3. Discussion

Our group previously identified 4464 circRNAs via the linear RNA-removed strategy-based deep sequencing of midgut tissues from *A. m. ligustica* worker bees [26]. Here, following the same protocol, 2083 circRNAs were identified for the first time in the gut tissues of *A. m. ligustica* larvae (Figure 1A). In view of the limited reports on *A. mellifera* circRNAs [17,26,27], the circRNAs discovered in this study could further enrich the reservoir of the circRNA data for Western honey bees and provide a valuable source for future research on circRNAs in *A. m. ligustica* and other subspecies belonging to *A. mellifera*. By comparison, we observed that 598 circRNAs were shared by *A. m. ligustica* larvae (gut tissue) and worker bees (midgut tissue) [26], which was in accordance with the tissue- and stage-specific expression characteristics of the circRNAs [28]. It is speculated that the total quantity of circRNAs in *A. m. ligustica* should be higher. In addition, we found that the most abundant circRNAs were exonic circRNAs (64.91% + 16.42%) and the least abundant were intronic circRNAs (Figure 1B), similar to the circRNAs identified in *A. c. cerana* [25], *B. mori* [19], and *L. striatellus* [20]. Moreover, the source genes of the identified circRNAs were distributed in all 16 chromosomes across the *A. mellifera* genome, ranging from 207 to 906 nt (Figure 1C). The distribution of circRNAs’ source genes in *A. mellifera* was consistent with that in other species such as chicken [29] and *A. thaliana* [30], indicating that this is a common property of circRNAs among diverse species.

In this work, there were 69, 92, and 100 unique circRNAs in the AmCK1, AmCK2, and AmCK3 groups, respectively, while the quantities of unique circRNAs in the AmT1, AmT2, and AmT3 groups were 118, 69, and 86, respectively. This was in line with the developmental stage- and stress stage-specific expression properties of the circRNAs. More efforts are needed to explore the functions of these common circRNAs. It could be inferred that the shared circRNAs in the larval guts were likely to exert fundamental functions during an *A. apis* infection, whereas the unique circRNAs probably played certain roles at specific time points. Intriguingly, we observed that there was no shared DEcircRNAs among these three comparison groups (Figure 2), suggesting that different circRNAs were differentially expressed at different stage of the *A. apis* infection. A different circRNA-adopted strategy may be a common phenomenon in *A. m. ligustica* and many other animals [31,32,33].

A major action mode of circRNA is to modulate the transcription of source genes [34,35]. Here, DEcircRNAs in the 4-, 5-, and 6-day-old comparison groups were involved in an array of crucial pathways including the metabolic pathway, phenylalanine metabolism, and several classical immune pathways such as the apoptosis, autophagy, endocytosis, as well as MAPK, Toll, and Imd signaling pathways. This indicated that corresponding DEcircRNAs probably exerted extensive regulatory functions via the modulation of the transcription of source genes, though the impact may be not strong, as shown in the *p* values for the enriched pathways (Figure 3D–F). For holometabolous insects, such as honey bees, storage proteins (e.g., HEX110, HEX70A, HEX70B, and HEX70C) not only provide the material basis for larval and pupal development but also play essential roles in various processes such as immunity and oviposition [36]. For example, Vieira et al. [37] found that miR-34 and miR-210 negatively regulated the expression of both *HEX70B* and *HEX110* by directly and redundantly binding to their 3′ untranslated region (UTR) sequences. In the present study, three source genes encoding storage proteins (HEX110, HEX70B, and HEX70C) were potentially targeted by 2, 3, and 27 DEcircRNAs in the 4-, 5-, and 6-day-old larval guts infected by *A. apis*, implying that these DEcircRNAs could affect the storage protein synthesis by regulating the transcription of the related source genes and further modulate the host immune defenses against an *A. apis* invasion.

The ecdysone receptor gene (*Ecr*) is considered to be a pivotal factor responsible for controlling the growth and development of arthropods [38]. Additionally, *Ecr* is one of the most important hormones in insects and acts in combination with juvenile hormones to regulate an array of processes, such as the growth as well as metamorphic and reproductive processes [39,40,41]. In this work, the *Ecr* gene was found to be a source gene of four DEcircRNAs, including novel_circ_001191 (log_2_(FC) = 18.80; *p* = 0.13), novel_circ_001192 (log_2_(FC) = 19.08; *p* = 0.13), novel_circ_001193 (log_2_(FC) = 17.80; *p* = 0.50), and novel_circ_001194 (log_2_(FC) = −2.69; *p* = 0.01). This demonstrated that these 4 DEcircRNAs could participate in the modulation of the immune defenses of *A. m. ligustica* larvae against an *A. apis* invasion in an *Ecr*-dependent manner.

A further analysis suggested that the DEmRNAs in the AmCK1 vs. AmT1, AmCK2 vs. AmT2, and AmCK3 vs. AmT3 comparison groups could be annotated to 24, 20, and 25 GO terms (Figure 5) and 62, 80, and 159 KEGG pathways (Figure 6), respectively. In total, there were a total of eight immune pathways enriched by DEmRNAs (ncbi_410860, ncbi_409006, ncbi_551500, etc.) (Appendix A), including the lysosome, endocytosis, phagosome, autophagy, apoptosis, as well as MAPK, Jak-STAT, Toll, and Imd signaling pathways (Figure 9).

An increasing amount of evidence has demonstrated that RNA molecules that contain MREs, including lncRNAs, circRNAs, and pseudogenes, are capable of competitively absorbing corresponding miRNAs to regulate the expression of downstream target genes [24,42,43]. In this work, we found that 21, 6, and 21 DEcircRNAs in the 4-, 5-, and 6-day-old larval guts could target 10, 3, and 7 DEmiRNAs, respectively, to form complex regulatory networks, as shown in Figure 4. This was suggestive of the potential of the DEcircRNAs to exert functions during *A. apis* infection via ceRNA networks. Previous studies have shown that miR-122 is a crucial regulator in a series of biological processes, such as autophagy, antioxidant defenses, and innate immunity [44]. For example, Liu et al. [45] observed that an *Aeromonas hydrophila* infection activated the expression of the cytokine *IL-15* in grouper fish by downregulating the miR-122 expression, which regulated the host immune responses. Here, a potential targeting relationship between novel_circ_001871 (log_2_(FC) = −19.59; *p* = 0.02) and miR-122-x was detected, suggestive of the involvement of the novel_circ_001871-miR-122-x axis in the host immune responses. However, additional work is required to clarify the underlying mechanisms. Recently, our group successfully established technical platforms for the functional investigation of both circRNAs [22] and miRNAs [46]. In the near future, we aim to conduct functional dissection of novel_circ_001871 and miR-122-x to unveil the mechanisms regulating the larval immune responses to an *A. apis* invasion.

## 4. Materials and Method

### 4.1. Fungi and Bee Larvae

*A. apis* spores were prepared using the method developed by Jensen et al. [47] and were then stored at −80 °C in the Honey Bee Protection Lab of the College of Animal Sciences (College of Bee Science), Fujian Agriculture and Forestry University, Fuzhou, China. *A. m. ligustica* larvae were gathered from three colonies that were reared in the apiary of the College of Animal Sciences (College of Bee Science), Fujian Agriculture and Forestry University, Fuzhou, China.

### 4.2. Source of the Strand-Specific cDNA Library-Based RNA-Seq Datasets

In our previous work, the gut tissues of *A. m. ligustica* worker larvae that had been inoculated with *A. apis* and the gut tissues of the corresponding uninoculated larvae were prepared and subjected to an RNA isolation and strand-specific cDNA library-based RNA-seq [48]. Briefly, (1) 3-day-old *A. m. ligustica* larvae (*n* = 48) were planted in a 48-well culture plate and were fed an artificial diet containing *A. apis* spores (the final concentration was 10^7^ spores/mL), while another set of 3-day-old larvae (*n* = 48) were planted in a separate 48-well culture plate and were fed an artificial diet without *A. apis* spores; (2) the larvae in the *A. apis*-inoculated group were reared in a chamber with a constant temperature (35 °C) and humidity (95%) (Shanghai Yiheng Scientific Instrument Co., Ltd., Shanghai, China), whereas those in the uninoculated group were reared in another chamber; (3) the guts of 4-, 5-, and 6-day-old *A. apis*-inoculated larvae (named the AmT1, AmT2, and AmT3 groups, respectively) and 4-, 5-, and 6-day-old uninoculated larvae (named the AmCK1, AmCK2, and AmCK3 groups, respectively) were dissected using our established protocol [49], producing a total of 9 gut tissue samples for each group; (4) the total RNAs in the gut samples from each group were extracted using the TRIzol method (Promega, Madison, WI, USA), subjected to cDNA synthesis using a QiaQuick PCR extraction kit (QIAGEN, Düsseldorf, Germany), and then sequenced on an Ilumina HiSeq^TM^ 4000 platform (Guangzhou Gene Denovo Biotechnology Co., Ltd., Guangzhou, China); (5) the produced raw data were subjected to a quality control to obtain high-quality clean reads, which were then used for the analyses performed in this study. The raw data were deposited in the NCBI Sequence Read Archive (SRA) database and linked to the BioProject number PRJNA408312. 

### 4.3. Source of the Small RNA-Seq Datasets

Using the prepared gut samples from the 4-, 5-, and 6-day-old *A. apis*-inoculated and uninoculated *A. m. ligustica* larvae, the RNA isolation, cDNA library construction, and sRNA-seq were conducted, followed by the quality control of the raw data to obtain high-quality clean tags [50], which were then used for our analyses. The raw data are available in the NCBI SRA database under the BioProject number PRJNA406998.

### 4.4. Characterization of the circRNAs

Using the method described by Chen et al. [26], the circRNAs were predicted using find-circ software [51] (https://github.com/orzechoj/circrna_finder, accessed on 22 November 2022) with default parameters. Subsequently, the predicted circRNAs were combined with the annotations from the *A. mellfera* reference genome (Amel_HAV3.1) to identify the different types of circRNAs. Further, the length distributions, cyclization types, and chromosome distributions of the circRNA source genes were calculated utilizing the find-circ software. Histograms were then drawn using the GraphPad Prism v8.0 software for Windows (San Diego, CA, USA).

### 4.5. Identification of the DEcircRNAs

The expression level of each circRNA was normalized to the FPKM. Based on the standards of *p* ≤ 0.05 (corrected by false discovery rate) and |log_2_(Fold change)| ≥ 1 (|log_2_(FC)| ≥ 1), the DEcircRNAs in each comparison group were screened. A Venn analysis of the DEcircRNAs in each comparison group was performed using the OmicShare platform (https://www.omicshare.com/ (accessed on 22 November 2022)).

### 4.6. Prediction and Investigation of the DEcircRNA Source Genes

According to our previously described protocol [25], the anchor reads at both ends of each circRNA were mapped to the *A. mellifera* reference genome (Amel_HAV3.1) using the Bowtie2 v2.3.4.2 software [52]. When both ends of one circRNA were aligned to the same gene, it was regarded as the source gene of that circRNA. Next, the source genes were annotated to GO (http://www.geneontology.org/ (accessed on 22 November 2022)) and KEGG (https://www.kegg.jp/ (accessed on 22 November 2022)) databases utilizing the OmicShare platform.

### 4.7. Construction and Analysis of the ceRNA Regulatory Networks

Using the previously described method by Chen et al. [53], the potential targeting relationships between the DEcircRNAs and DEmiRNAs, as well as those between the DEmiRNAs and DEmRNAs, were predicted using a combination of the MiRanda (V3.3a), RNAhybrid (V2.1.2) + SVM_light (V6.01), and TargetFind software [54] (http://targetfinder.org/, accessed on 22 November 2022). On the basis of the predicted targeting relationships, DEcircRNA-DEmiRNA-DEmRNA regulatory networks were constructed and then visualized using the Cytoscape v.3.2.1 software [55] with default parameters. Further, the targets were mapped to GO and KEGG databases and then plotted using the OmicShare platform.

### 4.8. PCR Amplification and Sanger Sequencing of the circRNAs

To confirm the authenticity of the circRNAs, 15 circRNAs were randomly selected for PCR amplification and Sanger sequencing, including novel_circ_000400, novel_circ_001048, novel_circ_001377, novel_circ_000408, novel_circ_000465, novel_circ_000573, novel_circ_000846, novel_circ_000922, novel_circ_001116, novel_circ_001374, novel_circ_001556, novel_circ_001620, novel_circ_000774, novel_circ_000344, and novel_circ_001194. The primers (as shown in Appendix A) across the back-splicing sites were designed using the Primer Premier 6 software [56] and synthesized using a Sangon Biotech (Shanghai, China) Co., Ltd. FastPure Cell/Tissue Total RNA Isolation Kit V2 (Vazyme, Nanjing, China), which was used to extract the total RNAs from the gut tissues of the 6-day-old *A. m. ligustica* larvae. Then, the linear RNAs were digested using RNase R (Geneseed, Guangzhou, China) to enrich the circRNAs. The cDNA of the circRNAs was obtained by reverse transcription using random primers and it was then used as a template for the PCR amplification, which was conducted on a T100 thermal cycler (NEB, Ipswich, MA, USA). The reaction system and procedure were set following the report by Chen et al. [26]. The amplified products were detected using 1.5% agarose gel electrophoresis with GoldView staining (Accurate, Beijing, China). Then, the target fragments were purified using a FastPure Gel DNA Extraction Mini Kit (Vazyme, Nanjing, China) and Sanger sequenced by Sangon Biotech (Shanghai) Co., Ltd. (Shanghai, China).

### 4.9. RT-qPCR Detection of the DEcircRNAs

The total RNAs were extracted from the gut samples from the three *A. apis*-inoculated groups and the three uninoculated groups using a FastPure Cell/Tissue Total RNA Isolation Kit V2 and were then divided into two portions: one portion was digested with RNase R to enrich the circRNAs and the resulting cDNA that was obtained by reverse transcription using random primers was used as the template for the RT-qPCR detection of the DEcircRNAs; the other portion was subjected to reverse transcription using Oligo dT primers and the resulting cDNA was used as the template for the RT-qPCR detection of the internal refence gene *actin* (GeneBank ID: 406122). The reaction was conducted on Applied Biosystems^®^ QuantStudio 3 (ABI, Waltham, MA, USA) under the following conditions: 94 °C pre-denaturation for 5 min; 94 °C denaturation for 50 s; 60 °C annealing and extension for 30 s; and a total of 36 cycles of qPCR reaction. There were three parallel samples, and the experiment was repeated three times. The reaction system (20 μL) contained 10 μL of SYBR Green dye, 1 μL of upstream and downstream primers (10 μmol/L), 1 μL of cDNA template, and 7 μL of DEPC water. The relative expression level of each DEcircRNA was calculated using the 2^−∆∆Ct^ method [57]. The data were shown as the mean ± standard deviation (SD) and subjected to Student’s *t*-test using the Graph Prism 8 software (ns *p* > 0.05; * *p* < 0.05; ** *p* < 0.01; *** *p* < 0.001). The details of the RT-qPCR primers that were used are presented in Appendix A.

## 5. Conclusions

In conclusion, 2083 circRNAs were identified for the first time in the guts of *A. m. ligustica* larvae. An *A. apis* infection caused overall changes in the expression profiles of the circRNAs in the host guts. Additionally, the DEcircRNAs putatively participated in the immune responses of the larvae to an *A. apis* invasion by directly regulating the expression of source genes or indirectly modulating the downstream gene expression via interactions with the target DEmiRNAs. Furthermore, the corresponding DEcircRNAs were potentially engaged in host immune responses through ceRNA regulatory networks via the absorption of miR-122-x.

## Figures and Tables

**Figure 1 ijms-24-00613-f001:**
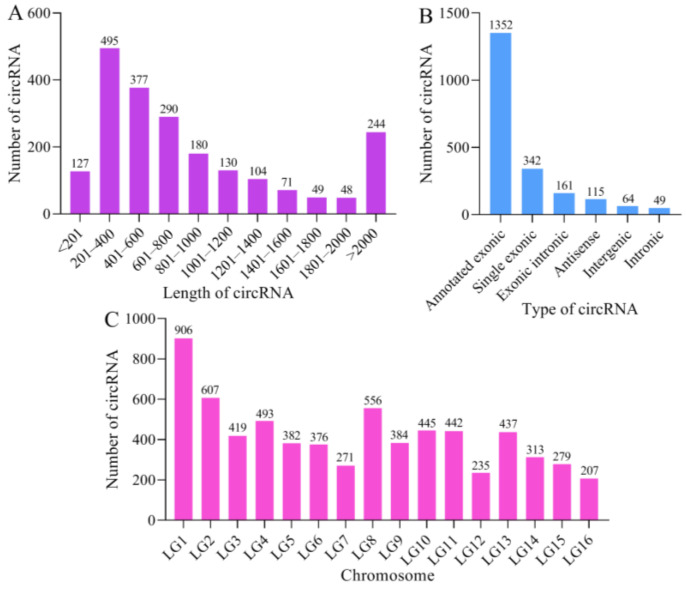
Characteristics of the identified *A. m. ligustica* circRNAs: (**A**) length distribution of circRNAs; (**B**) type distribution of circRNAs; (**C**) chromosomes enriched by circRNAs.

**Figure 2 ijms-24-00613-f002:**
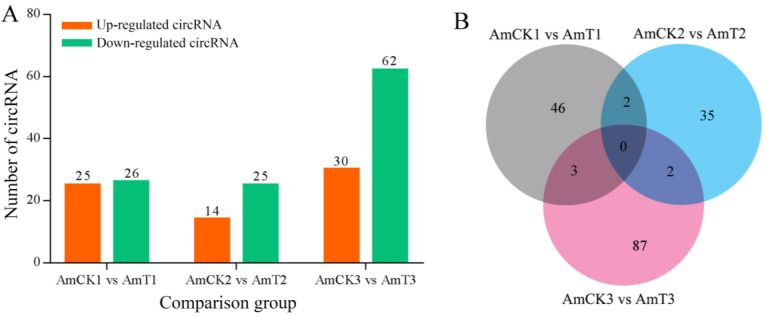
Number statistics (**A**) and Venn diagram (**B**) of DEcircRNAs in three comparison groups.

**Figure 3 ijms-24-00613-f003:**
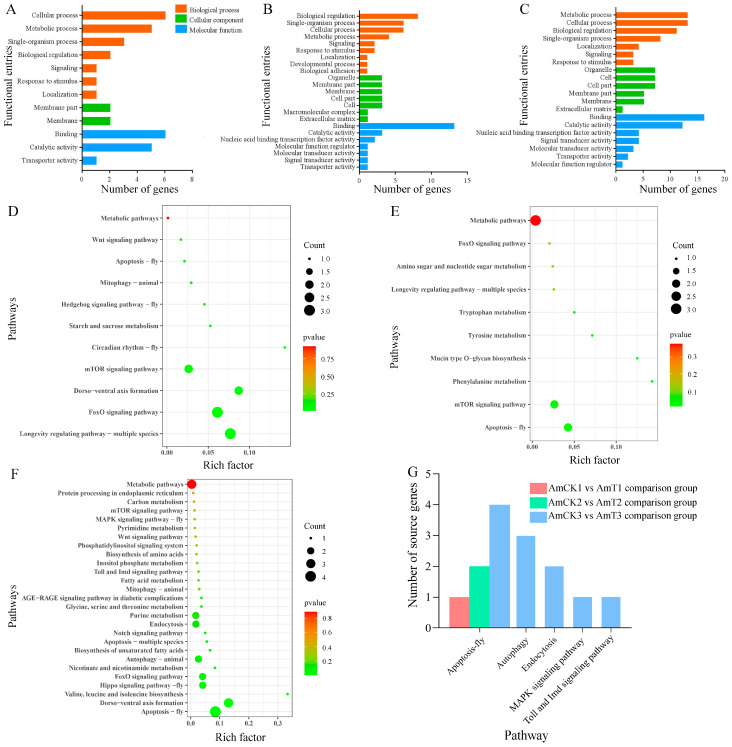
Annotations of the source genes of the DEcircRNAs in the three comparison groups: (**A**–**C**) GO database annotations of the source genes; (**D**–**F**) KEGG database annotations of the source genes; (**G**) number statistics of the source genes related to cellular and humoral immune pathways.

**Figure 4 ijms-24-00613-f004:**
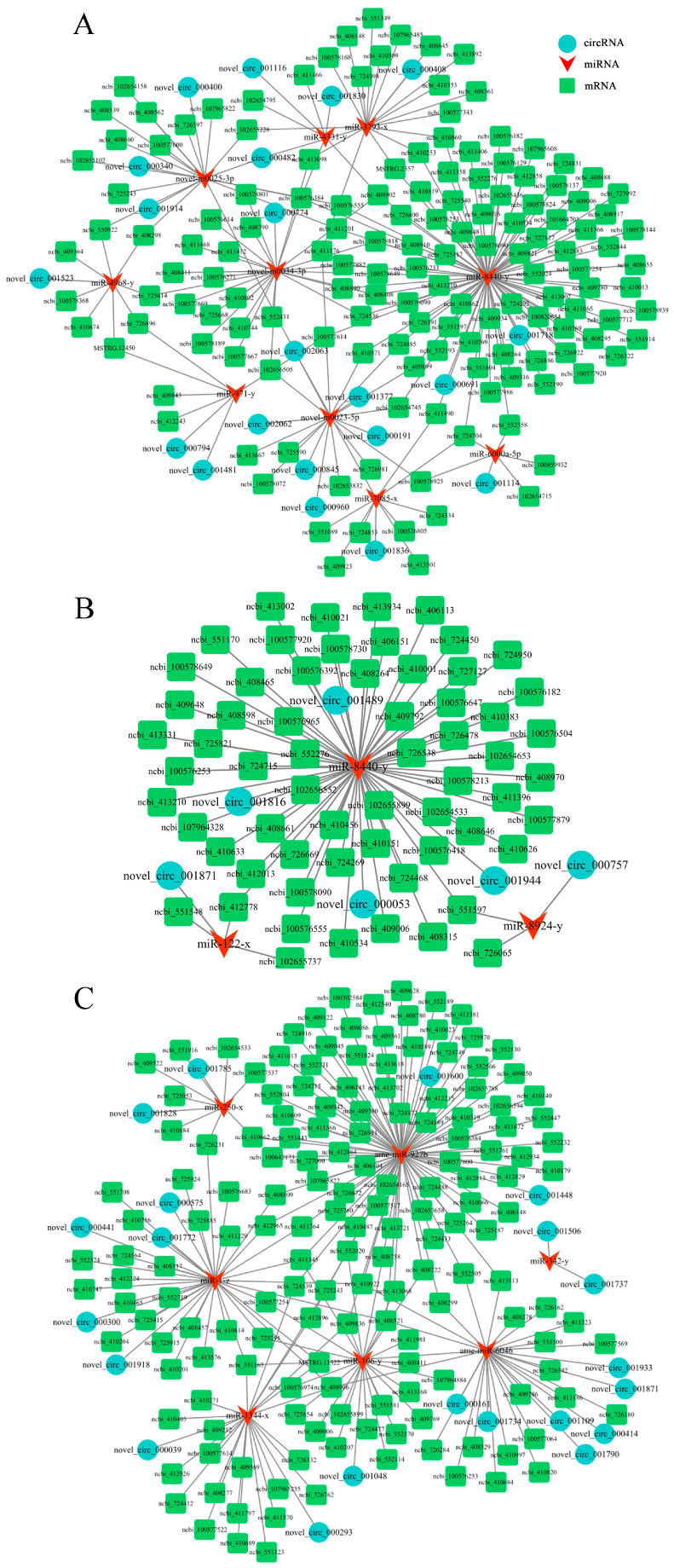
DEcircRNA–DEmiRNA–DEmRNA networks in (**A**) the AmCK1 vs. AmT1, (**B**) AmCK2 vs. AmT2, and (**C**) AmCK3 vs. AmT3 comparison groups.

**Figure 5 ijms-24-00613-f005:**
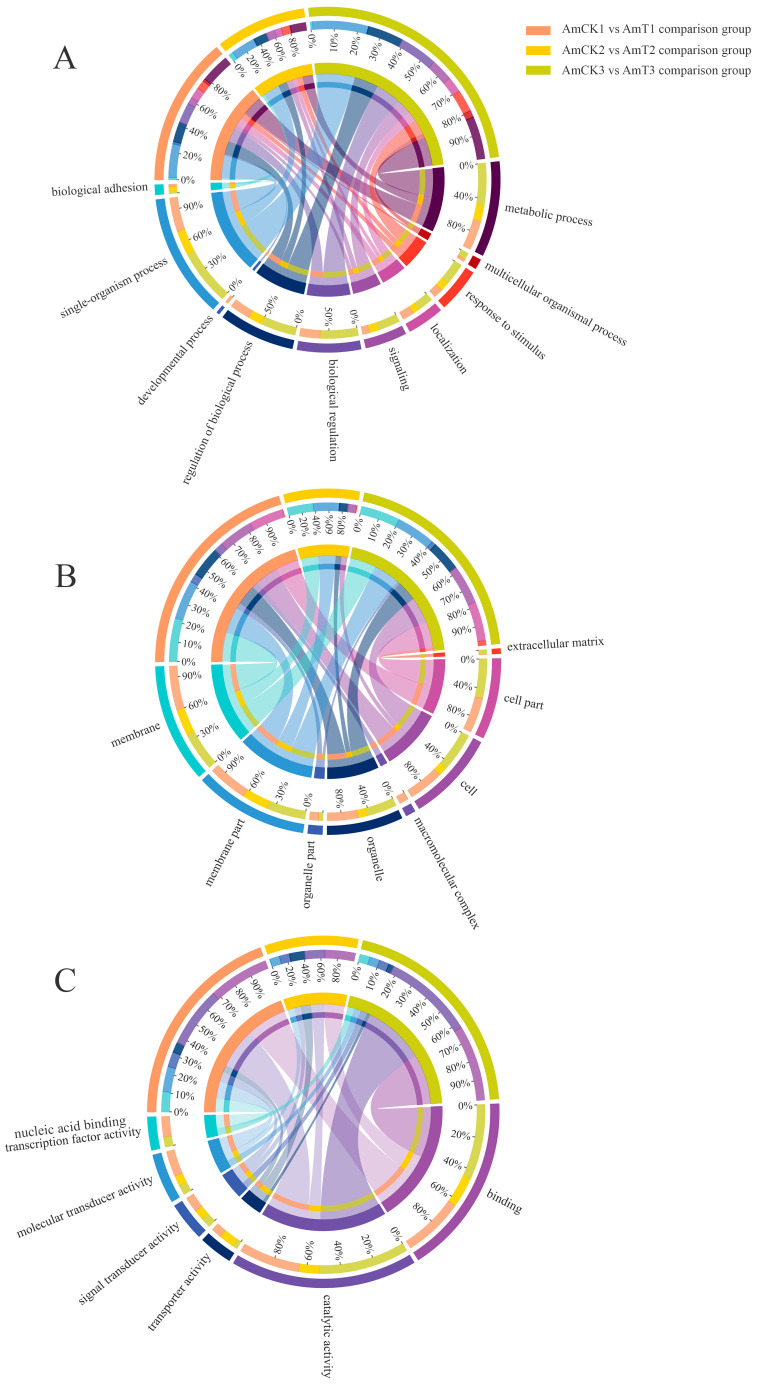
GO terms annotated by the target DEmRNAs in the three comparison groups: (**A**) the annotated terms regarding biological processes; (**B**) the annotated terms regarding cellular components; (**C**) the annotated terms regarding molecular functions. Different colors represent different GO terms. The percentages indicate the proportions of the source genes annotated to the same GO term in all source genes targeted by DEcircRNAs in every comparison groups.

**Figure 6 ijms-24-00613-f006:**
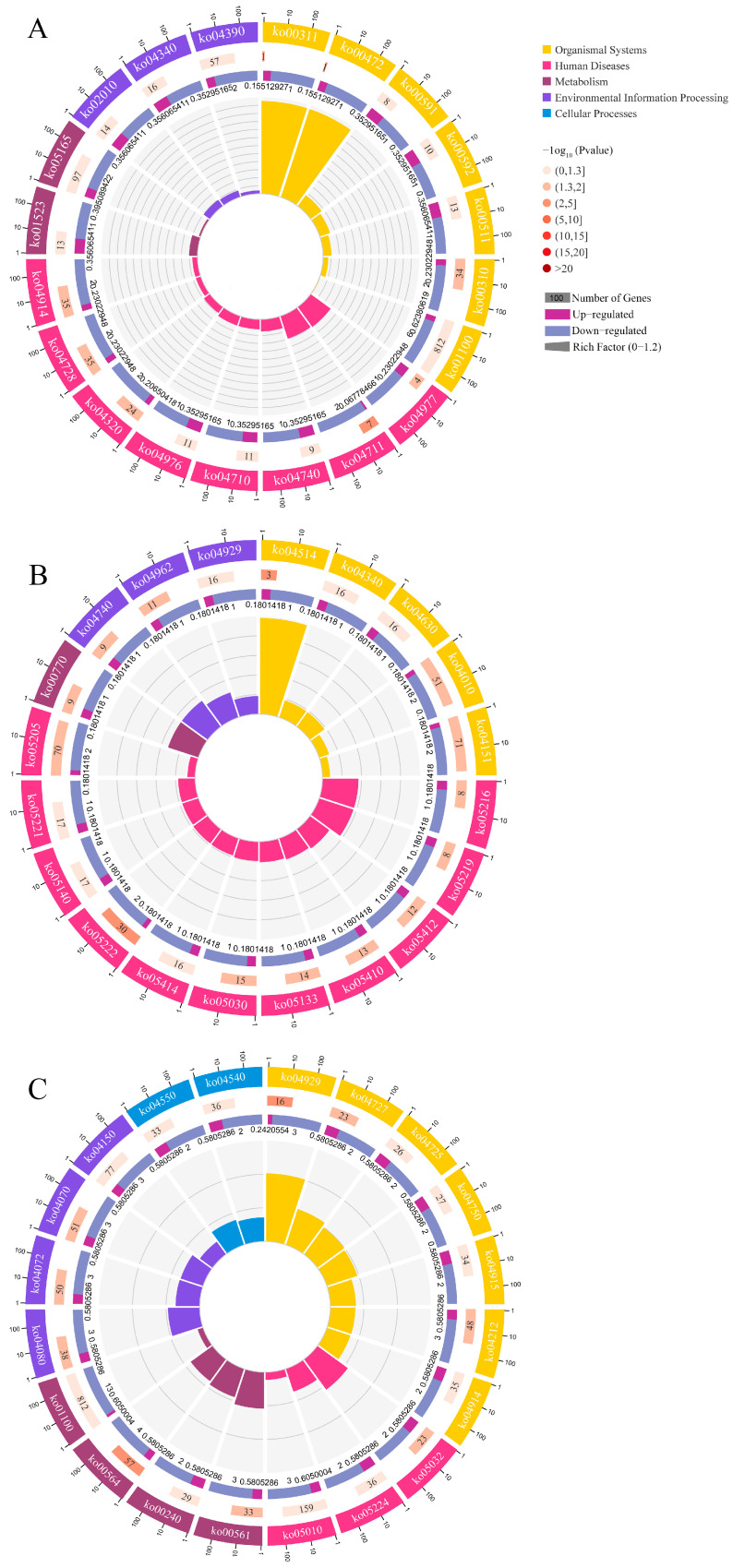
KEGG pathways enriched by targets in the (**A**) AmCK1 vs. AmT1, (**B**) AmCK2 vs. AmT2, and (**C**) AmCK3 vs. AmT3 comparison groups.

**Figure 7 ijms-24-00613-f007:**
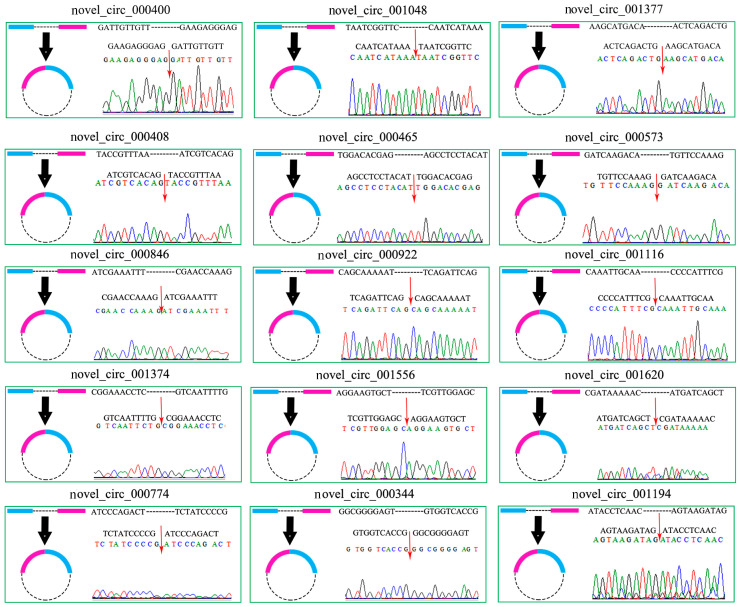
The Sanger sequencing of the amplified fragments from 15 *A. m. ligustica* circRNAs. The red arrows indicate the back-splicing sites.

**Figure 8 ijms-24-00613-f008:**
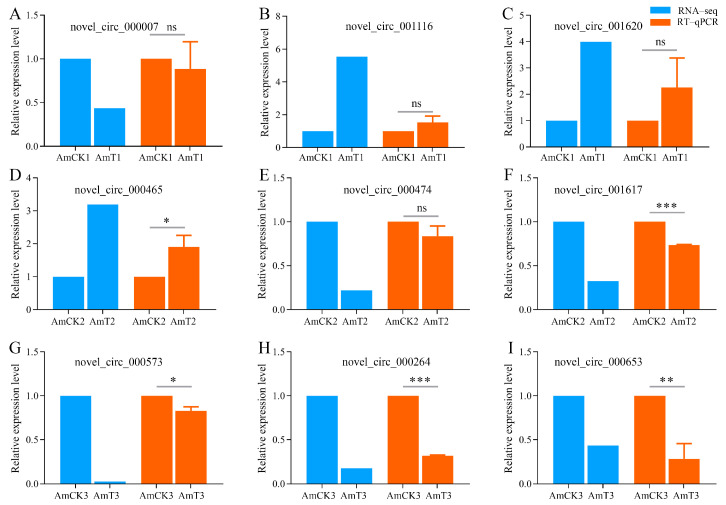
RT-qPCR validation of 9 DEcircRNAs. ns, *p* > 0.05; * *p* < 0.05; ** *p* < 0.01; *** *p* < 0.001. (**A**–**C**) Three DElncRNAs in AmCK1 vs AmT1 comparison group; (**D**–**F**) Three DElncRNAs in AmCK2 vs AmT2 comparison group; (**G**–**I**) Three DElncRNAs in AmCK3 vs AmT3 comparison group.

**Figure 9 ijms-24-00613-f009:**
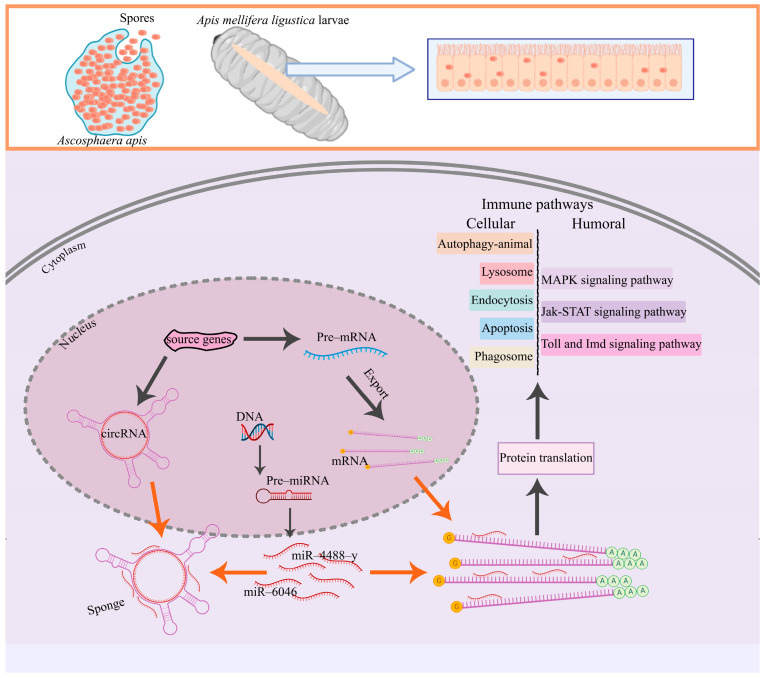
A hypothetical schematic diagram of circRNA-mediated immune responses of western honey bee larvae to *A. apis* invasion. The blue arrow indicates the enlarged view of larval epithelial cells; the orange arrows indicate the competitive binding of miRNAs between DEcircRNAs and DEmRNAs; the black arrows indicate the transcription of source genes and the translation of source gene-derived DEmRNAs as well as downstream cellular and humoral immune responses.

## Data Availability

The data supporting reported results can be found in biorxiv website (https://www.biorxiv.org/ (accessed on 22 November 2022)).

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
