# Peer review of "Unveiling the circRNA-Mediated Immune Responses of Western Honey Bee Larvae to Ascosphaera apis Invasion"

_ijms, 2022, doi:10.3390/ijms24010613_

Round 1

Reviewer 1 Report

Ye and coauthors reported the response of cricRNAs to the fungus infection. The background introduction is sufficient. The results and discussion are well  presented. I have two minor comments:

In Figure 3, the P value for the enriched pathways was quite high. Does it mean that the affects on the pathway regulation is not strong?

In figure 5 It is not clear what does the number and color mean.

Author Response

Dear Reviewers,

Thanks for your comments and recommendations of great importance, which significantly improved the quality of our work. We seriously examined the whole manuscript and made corresponding modifications. All changes have been shown in “Track changes” in the revised manuscript.

Response to Reviewer 1:

  1. In Figure 3, the Pvalue for the enriched pathways was quite high. Does it means that the affects on the pathway regulation is not strong?

Response: Thank you so much for your helpful comment. circRNA can exert regulatory functions by diverse manners, such as regulation of the source gene’s transcription and modulation of downstream gene expression via ceRNA network. Figure 3 showed the annotated pathways by source genes of DEcircRNAs in the Apis mellifera ligustica larval guts infected by Ascosphaera apis. As you said, the P value for the enriched pathways was quite high, indicating that DEcircRNAs may affect these pathways through modulation of the transcription of source genes though the impact was not strong. Accordingly, we improved the related content in the discussion section following your comment.

  1. In figure 5 It is not clear what does the number and color mean.

Response: Thanks for your valuable comment, based on which we added necessary descriptions in the figure legend to give more detailed information. Please see the revised version of manuscript.

Reviewer 2 Report

I would appreciate author’s work about the circRNA-mediated immune responses of western honeybee larvae to Ascosphaera apis invasion. However, there are some suggestions the authors need to pay attention to before the manuscript is acceptable for publication:

1.Line 129-130. Why there were no overlaped DEcircRNAs among these 3 groups? How to explain the phenomenon?

2.This question is about analysis method in the manuscript. Why not compare AmT1, AmT2 and AmT3 in order to explore the DEcircRNAs among different days after inoculated?

3.This question is about writing in words. Please check up the number under 10, I suggest that they need to be changed into Roman numerals, e,g, in Line 160-167.

4.Line 151. MAPK pathway is a common pathway under external stress. Did author find some more interesting pathways and related DEcircRNAs which would indicate the biological changes after the treatment?

5.Line 200-202. Compared with RNA-seq and RT-qPCR, panel A, B, C and E were shown that there was no significant difference between treatment groups and control groups. Why said that their trends were the same? What was the author's standard of “the same trends”?

6.Line 281. Figure 9 shown a schematic diagram of circRNA-mediated immune responses of western honeybee larvae to A. apis invasion. Do authors have some solid experimental evidence that can support the model?

Author Response

Dear Reviewers,

Thanks for your comments and recommendations of great importance, which significantly improved the quality of our work. We seriously examined the whole manuscript and made corresponding modifications. All changes have been shown in “Track changes” in the revised manuscript.

Response to Reviewer 2:

  1. Line 129-130. Why there were no overlaped DEcircRNAs among these 3 groups? How to explain the phenomenon?

Response: Thanks for your valuable comment. In this work, Venn analysis showed that there was no shared DEcircRNAs among three comparison groups. Given that circRNAs were suggested to have developmental- and stress stage-specific expression characteristics, we speculated that different circRNAs were differentially expressed at different stage of A. apis infection. Additionally, we checked plenty of associated studies with circRNAs following your helpful comment, and found that different circRNA-adopted strategy was also a common phenomenon in other animals (Haddad, G.; Lorenzen, J. M. Biogenesis and Function of Circular RNAs in Health and in Disease. Front Pharmacol 2019, 10, 428; Memczak, S.; Jens, M.; Elefsinioti, A.; Torti, F.; Krueger, J.; Rybak, A.; Maier, L.; Mackowiak, S. D.; Gregersen, L. H.; Munschauer, M.; et al. Circular RNAs are a large class of animal RNAs with regulatory potency. Nature 2013, 495(7441), 333-8; Zhang, Z.; Yang, T.; Xiao, J. Circular RNAs: Promising Biomarkers for Human Diseases. EBioMedicine vol. 2018, 34, 267-274.). Accordingly, we improved the discussion section by adding some new contents.

  1. This question is about analysis method in the manuscript. Why not compare AmT1, AmT2 and AmT3 in order to explore the DEcircRNAs among different days after inoculated?

Response: In fact, the major objective of this study was to investigate the expression pattern, regulatory network, and potential function of circRNAs in the A. m. ligustica larval guts responding to A. apis infection. If we explore the DEcircRNAs by direct comparison between AmT1 and AmT2 groups and between AmT2 and AmT3 groups, it’s hard to eliminate those DEcircRNAs during the developmental process (4- to 6-day-old) of the larval guts. For example, by comparing AmT1 and AmT2 groups, we can screen some DEcircRNAs, which contained not only DEcircRNAs due to A. apis infection but also DEcircRNA due to the development of larval gut. Hence, we compared the un-inoculated and A. apis-inoculated groups at each day-old to explore DEcircRNAs, which were believed to be directly related to host response to A. apis infection. Thanks.

  1. This question is about writing in words. Please check up the number under 10, I suggest that they need to be changed into Roman numerals, e,g, in Line 160-167.

Response: Following your helpful recommendation, we carefully checked the numbers throughout the manuscript and made necessary modifications. Thanks.

  1. Line 151. MAPK pathway is a common pathway under external stress. Did author find some more interesting pathways and related DEcircRNAs which would indicate the biological changes after the treatment?

Response: Following your kind comment, we reexamined the total 27 pathways enriched by source genes of DEcircRNAs in the AmCK3 vs. AmT3 comparison group, especially the six immune pathways since the major objective of this work is to analyze host immune responses regulated by DEcircRNAs. Accordingly, we seriously modified the related description in the results section in the revised version of manuscript. Thanks.

  1. Line 200-202. Compared with RNA-seq and RT-qPCR, panel A, B, C and E were shown that there was no significant difference between treatment groups and control groups. Why said that their trends were the same? What was the author's standard of “the same trends”?

Response: Thanks. For the sake of validation of the reliability of the transcriptome date used in this work, RT-qPCR was conducted to detect the expression trends of nine randomly selected DEcircRNAs, as Figure 8 A,B,C,E shown, the differences between treatment groups and control groups were not significant but roughly the same as the differences between treatment groups and control groups in the sequencing datasets. “the same trends” described here was inaccurate. Following your kind comment, we carefully modified the corresponding descriptions in the revised manuscript.

  1. Line 281. Figure 9 shown a schematic diagram of circRNA-mediated immune responses of western honeybee larvae to A. apisinvasion. Do authors have some solid experimental evidence that can support the model?

Response: Thanks for your valuable comment. In fact, we mainly conducted related analyses based on the obtained high-quality transcriptome data and bioinformatics, followed by necessary molecular validation of the back-splicing sites within some circRNAs and expression trends of some DEcircRNAs. On basis of the findings in this work and your helpful comments as well as suggestions, we plan to perform functional investigation of key DEcircRNAs (eg. those DEcircRNAs within ceRNA networks and had the highest connectivity) in the near future. After serious thinking, the title of Figure 9 was replaced with “A hypothetical schematic diagram of circRNA-mediated immune responses of western honey bee larvae to A. apis invasion”.

Round 2

Reviewer 2 Report

The authors have corrected all the mistakes and updated the writing. Now, from my point of view, it is acceptable for publication in IJMS.